# Microbiome of Free-Living Amoebae (FLA) Isolated from Fresh Organic Produce: Potential Risk to Consumers?

**DOI:** 10.3390/foods12163102

**Published:** 2023-08-18

**Authors:** Lara Soler, Yolanda Moreno, Laura Moreno-Mesonero, Inmaculada Amorós, José Luís Alonso, María Antonia Ferrús

**Affiliations:** 1Research Institute of Water and Environmental Engineering (IIAMA), Universitat Politècnica de València, Camino de Vera s/n, 46022 Valencia, Spain; larasolergarcia@gmail.com (L.S.); laumome@upv.es (L.M.-M.); iamoros@ihdr.upv.es (I.A.); jalonso@ihdr.upv.es (J.L.A.); 2Biotechnology Department, Universitat Politècnica de València, Camino de Vera s/n, 46022 Valencia, Spain; mferrus@btc.upv.es

**Keywords:** free-living amoebae, amoeba-resistant bacteria, microbiome, organic vegetables

## Abstract

In response to growing global interest in organic agriculture, this study delves into the microbial landscape of organically grown raw produce with a focus on food safety. Vegetables that are consumed raw are potential vehicles for the transmission of any type of microorganism capable of causing human disease. Free-living amoebae (FLA) are ubiquitous protozoa found in many ecosystems and can serve as hosts to pathogenic bacteria. So far, data regarding the FLA bacterial microbiome in fresh produce remain scarce and are non-existent for those of organic origin. Thus, the aim of this preliminary work is to characterize the microbiome of FLA in commonly consumed raw vegetables to know their possible implications for consumers. A total of 40 organic cabbage, lettuce, spinach, and strawberry samples were analyzed. FLA were found in all samples, and their bacterial microbiome was obtained via amplicon sequencing using the Illumina MiSeq platform and pair-end protocol. *Acanthamoeba* spp. and *Vermamoeba vermiformis* were identified via qPCR in 65.0% and 25.0% of the samples, respectively. Regarding the bacterial microbiome of FLA, the most abundant genera were *Pseudomonas* (1.8–17.8%) and *Flavobacterium* (1.7–12.6%). Bacteria not previously related to FLA, such as *Prosthecobacter* or *Cellvibrio*, are described in this work. Importantly, several bacterial genera found within the FLA microbiome were identified as potential human pathogens, including *Pseudomonas*, *Flavobacterium*, *Arcobacter*, *Klebsiella*, *Mycobacterium*, *Salmonella* and *Legionella*. This is the first work in which FLA microbiome isolated from organic products has been characterized, underscoring the significance of understanding FLA’s role as carriers of pathogenic bacteria in the context of organic food safety.

## 1. Introduction

There is currently a growing global interest in organic agricultural production in search of sustainability and the improvement of public health. In addition, interest in organic food has grown markedly as consumers and merchants react to the effects of pesticides on human health, the environment, and food safety [1].

In other words, due to environmental awareness, health concerns, interest in improving quality of life, safety, and ethical and political reasons, the population consumes plant-based organic products [2,3]. These are produced from organic agriculture, a production system that promotes biodiversity and the biological cycle of the soil, crops and livestock [4]. In comparison, conventional methods use synthetic pesticides and fertilizers, which cause infertility in the soil and can be carcinogenic to humans [3].

In Spain, as in most developed countries, the most advanced organic primary production sector is that of vegetable origin. In the same way, both fruits and vegetables are the most consumed organic foods [5]. Vegetables that are consumed raw are potential vehicles for the transmission of any type of pathogen [6]. However, it should be noted that there is a knowledge gap in some aspects of the microbiological quality and safety of organic food [5]. The main risk of organic food consumption lies in bacterial contamination.

Some established sources of microbiological contamination of fresh produce include soil, livestock, wildlife, contaminated manure, and the use of fertilizers or irrigation, the last two being the main risk factors [7]. Pathogens which are found in soil can also contaminate crops via direct contact with contaminated manure, vectors, heavy rains or spray watering [8].

Irrigation water has been identified as an important reservoir and a potential vehicle for foodborne pathogens, as these can remain viable and survive for several months. Indeed, the relationship between some parasites, such as protozoa, belonging to the genera *Cryptosporidium* or *Giardia*, which are present in water, and the contamination of vegetables has been shown [9].

Free-living protozoa such as free-living amoebae (FLA) are ubiquitous in many ecosystems such as surface water, wastewater, or drinking water as bacterial predators [10]. However, some bacteria can survive within FLA and then would also be able to survive immune cell phagocytosis [11]. Although some FLA species can act as hosts to pathogenic bacteria (the so-called “Trojan horses”) that could cause a threat to public health [12], there is little information about their presence in vegetables. These are often not included in microbiological surveillance, as they wrongly tend to be considered harmless microorganisms. Consequently, data regarding FLA in food remain scarce and, as far as we are concerned, there is only one previous work in which the presence of FLA in lettuce and the subsequent analysis of their bacterial microbiome was determined so far, thus suggesting that FLA are potential carriers of human pathogens [13].

Available studies have mainly focused on isolating FLA from a wide variety of vegetables such as lettuce, spinach, and vegetable sprouts [14,15]. FLA isolated from vegetable sprouts were found to have *Acanthamoeba* and *Vannella* as the dominant taxa. Moreover, FLA proved to be resistant to vegetable treatments such as washing and sanitation since they were also recovered from ready-to-eat lettuce [15].

Although a wide range of articles describe the interactions of FLA with amoeba-resistant bacteria (ARB), very few have studied the complete bacterial microbiome of FLA [16]. So far, only three studies have addressed this issue using next-generation sequencing (NGS) technologies, determining the microbiome of FLA isolated from drinking water sources [17], wastewater [18], and vegetables [13]. Therefore, in order to establish which bacteria tend to interact with FLA and to discover new potential ARB, further research on the bacterial microbiome of FLA is necessary.

From a public health perspective, the presence of pathogenic bacteria in FLA is of concern for several reasons. First, FLA act as hosts for foodborne pathogenic bacteria such as *Campylobacter jejuni*, *Legionella pneumophila*, or *Listeria monocytogenes*, among others, to which they offer protection, ensuring their survival [19]. Second, these internalized bacteria are able to develop escape routes and overcome cellular digestion mechanisms [20]. Moreover, some FLA are pathogenic themselves [21].

For this reason, in this study, a preliminary characterization of the bacterial microbiome of FLA isolated from organically grown fresh produce which are frequently eaten raw was carried out to know their implications for consumers’ health.

## 2. Materials and Methods

### 2.1. Samples and Processing

A total of 40 samples of fresh organic produce which are typically consumed raw were purchased from different local stores in the city of Valencia (Spain) from November 2020 to December 2021. The samples included 11 cabbage (*Brassica oleracea* var. *capitata*), 11 lettuce (*Lactuca sativa*), 11 spinach (*Spinacia oleracea*), and 7 strawberry (*Fragaria ananassa*) samples.

Fifty grams per sample (outer leaves of vegetable samples) were placed in a Stomacher sterile bag (REF.252019; Interscience, France) containing 250 mL of PAS buffer (Page’s Amoeba Saline; 1 g/L sodium citrate, 0.4 mM CaCl_2_, 4 mM MgSO_4_, 2.5 mM Na_2_HPO_4_, 2.5 mM KH_2_PO_4_, 50 μM Fe(NH_4_)_2_(SO_4_)_2_, pH 6.5). Cabbage, lettuce, and spinach samples were homogenized in a mixer (Homogenius HG 400, MAYO International, Italy) for 1 min at medium speed. Strawberry samples were gently homogenized by hand for 1 min.

Thereafter, homogenates without organic residuals were filtered through nitrocellulose filters with a pore size of 1.2 μm. Filters were then aseptically placed upside-down into non-nutrient agar (NNA; bacteriological agar dissolved in PAS (2 g/100 mL)) plates, and incubated at 28 °C. After 24 h, the filters were removed, and the plates continued at 28 °C for a week or until FLA growth was observed, for a maximum period of 30 days.

### 2.2. FLA Isolation

FLA growth was checked using a phase contrast microscope (40×). The contents of FLA-positive plates were recovered by adding PAS buffer to NNA plates and using a sterile cell lifter. The contents were concentrated via centrifugation (3000 rpm for 8 min) and resuspended in PBS buffer. Afterwards, sodium hypochlorite was added at a final concentration of 100 ppm to make sure to kill all bacteria non-internalized into FLA. The solution was kept in darkness for an hour at room temperature. Thereafter, sodium hypochlorite was eliminated via centrifugation (3000 rpm for 8 min), and the sediment was resuspended in 500 μL of PBS and subsequently treated with propidium monoazide (PMA; Biotium, Inc., Fremont, CA, USA) at a final concentration of 50 µM. PMA was used to eliminate dead cells and free DNA fragments present outside the FLA as previously explained [13]. Then, samples were frozen at −20 °C until DNA purification was performed.

### 2.3. DNA Purification and Amplicon Sequencing

DNA was purified using the GeneJET™ Genomic DNA Purification Kit (Thermo Scientific, Bremen, Germany), following the manufacturer’s instructions for cultured mammalian cells, although the incubation period at 56 °C was extended until 30 min as previously performed [22]. Amplicon sequencing was performed in Illumina MiSeq via a 2 × 300 bp paired-end run at FISABIO sequencing service (Valencia, Spain). Amplicon libraries were generated using the recommended set of primers and conditions specified by the 16S Metagenomic Sequencing Library Preparation guide, which target the 16S rRNA V3-V4 region and creates a single amplicon of approximately 460 bp (Part # 15044223 Rev. B).

### 2.4. Data Analysis

Raw data was analyzed using Quantitative Insights into Microbial Ecology 2 (QIIME2) v2020.11 software (https://qiime2.org/ (accessed on 15 June 2023)) [23]. Sequences were first imported. Then, forward and reverse sequences were merged, quality and chimera-checked, and clustered into Amplicon Sequence Variants (ASVs) defined at 99% sequence similarity using the DADA2 algorithm [24]. Sequences with a length of less than 200 bp and with a minimum phred Q30 quality value were discarded. Thereafter, ASVs underwent taxonomic assignment using the “classify-consensus-blast” plugin [25] and SILVA SSU v138.1 database [26]. Subsequently, ASVs assigned to chloroplasts and mitochondria were filtered out. Then, samples were rarefied to 38,287 sequences per sample so that comparisons among them could be performed. Alpha-diversity was evaluated using observed features and Faith’s Phylogenetic Diversity indices [27]. Beta diversity on both unweighted and weighted UniFrac [28] were estimated using q2-diversity after samples were rarefied (subsampled without replacement) and visualized using principal coordinate analysis (PCoA). Beta diversity analysis was computed to establish the differences between microbial communities between the distinct types of fresh organic produce (cabbage, lettuce, spinach, or strawberry). Moreover, ASVs of the genera containing pathogenic species were aligned against BLAST database using the blastn suite (https://blast.ncbi.nlm.nih.gov/Blast.cgi (accessed on 16 June 2023)). Plots were generated using the ggplot2 package [29] in R Software v4.2.2 [30].

### 2.5. Acanthamoeba spp. and Vermamoeba vermiformis qPCR Identification

*Acanthamoeba* spp. identification was conducted using the primers AcantF900 and AcantR1100 to amplify a 180 bp 18S rRNA fragment and the TaqMan probe AcantP1000 [31]. TaqMan qPCR reactions were performed in a final volume of 20 µL which contained 0.8 µL of each primer (10 µM), 0.4 µL of the probe (10 µM), 4 µL of LightCycler^®^ TaqMan^®^ Master Reaction mix (Roche Applied Science, Barcelona, Spain), 0.4 µL of BSA (1 mg/mL), and 3 µL of DNA template. Reactions were run in a LightCycler 2.0 with an initial DNA denaturation step at 95 °C for 10 min, followed by 40 cycles of: 95 °C for 10 s, 63 °C for 8 s and 72 °C for 7 s. A positive control with *Acanthamoeba castellanii* DNA and a negative control without DNA were included in all assays. Amplifications were conducted in duplicate.

*Vermamoeba vermiformis* identification was conducted using the primers Hv1227F and Hv1728R, which amplify a 502 bp 18S rRNA fragment [10]. SYBR Green qPCR was performed in a final volume of 20 µL which contained 0.4 µL of each primer (10 µM), 2 µL of LightCycler^®^ FastStart DNA Master SYBR Green I (Roche Applied Science, Spain), 2.4 µL of MgCl_2_, 2 µL of BSA (4 mg/mL) and 4 µL of DNA template. The cycling conditions conducted in LightCycler 2.0 were an initial denaturation step at 95 °C for 10 min, followed by 40 cycles of: 95 °C for 10 s, 56 °C for 10 s and 72 °C for 25 s. A positive control with *V. vermiformis* DNA and a negative control without DNA were included in all assays. Amplifications were also conducted in duplicate. The characteristic melting temperature (Tm) for *V. vermiformis* amplicon was 88.3 ± 0.6 °C.

Standard curves for each qPCR were constructed with tenfold serial dilutions of DNA from each protozoan. Triplicate analyses were run for each DNA dilution.

## 3. Results and Discussion

With the aim of contributing crucial information to the field of food safety by shedding light on the microbial dynamics in organically grown raw vegetables and their implications for public health, characterization of FLA bacterial microbiome from organic raw produces have been analyzed in this work.

Few studies have reported the presence of FLA in neither vegetables nor fruit so far. Our group has previously identified the presence of FLA in lettuce samples (100%), but in that case, vegetables were conventionally grown [13]. Similarly, in the present work, FLA growth was observed in all processed fresh organic produce (40/40, 100%), this being the first study in which organic vegetables have been checked for the presence of FLA. Moreover, other authors studied the presence of free-living protozoa, a group in which FLA are included, and all samples contained this group of microorganisms [15]. These authors concluded that washing or rinsing lettuce leaves followed by spin-drying in a household salad spinner reduced the protozoan numbers only by a maximum 1 log unit. They concluded that it is currently unknown to what extent industrial washing and sanitation affect FLA populations on produce surfaces, probably because they are very resistant to different stress conditions in their cyst form. Chavatte et al. [14] analyzed the presence of FLA in vegetable sprouts, finding them in 79% of the samples. In this case, *Acanthamoeba* spp. and *Vannella* spp. were the dominant genera. In addition, other authors studied the prevalence of protozoa in lettuce and spinach samples [32]. However, in contrast with our results, a low presence of FLA was reported. These authors concluded that FLA presence was underrepresented in their work, probably because the outer leaves were trimmed off the vegetables prior purchasing them, thus leaving the most-colonized leaves out of the analysis. In this sense, in the current work, as it has been mentioned, the outer leaves of the organic vegetable samples were selected for the analyses to recover the most diversity of the FLA present in the samples.

More recently, the presence of *Acanthamoeba* spp. in different fresh vegetables (garden cress, chives, mint, parsley, and basil) has been studied [33]. In this study, 25.7% of the vegetable samples were positive for *Acanthamoeba* spp. growth based on morphology criteria. In the present study, the presence of both *Acanthamoeba* spp. and *V. vermiformis* has been evaluated via qPCR. Results indicated the occurrence of *Acanthamoeba* spp. and *V. vermiformis* in 65.0% and 25.0% of the samples, respectively (Table 1). To our knowledge there are no previous reports about the identification of these FLA in vegetables or fruits, neither conventional nor organic using qPCR technique.

To elucidate the microbiome of FLA isolated from organic cabbage, lettuce, spinach, and strawberry samples, demonstrating their role as Trojan horses for potential pathogenic bacteria, 16S rRNA amplicon sequencing technique was applied after a treatment to kill non-internalized bacteria and block their DNA from amplification. After amplicon sequencing with Illumina MiSeq platform, 5,708,286 raw reads were obtained. A total of 3,386,019 reads remained after quality filtering, chimera removal and chloroplasts and mitochondria filtering out. Samples’ reads were rarefied to 38,287 reads per sample to compare them. Remaining reads were clustered into 9625 ASVs.

The most abundant bacterial phyla were Proteobacteria and Bacteroidota (also known as Bacteroidetes), which both represented an average of 80.84% of the bacterial microbiome of FLA among the samples (Figure 1A). The phylum Proteobacteria was the most abundant in all types of samples (i.e., cabbage, lettuce, spinach, and strawberry). Similarly, Bacteroidota was the second most abundant phylum in all cases. However, the third most abundant phylum differed in each type of fresh produce, Myxococcota being the third most abundant phylum in organic lettuce, Verrucomicrobiota in both organic spinach and cabbage samples, and Firmicutes in strawberry samples (Figure 1A, Appendix A). Moreno-Mesonero et al. [13] studied the microbiome of 20 conventionally grown lettuce samples and found that the most represented phyla were Proteobacteria and Bacteroidota, thus coinciding with the results obtained in the present study at phylum level. So far, this is the first publication which has reported on the study of FLA microbiome from fresh organic produce. Other authors also characterized FLA microbiome via amplicon sequencing, but in those cases, FLA were isolated from water sources [17,18].

The most abundant classes of bacteria associated with FLA in the present study were Gammaproteobacteria, Bacteroidia, Alphaproteobacteria and Verrucomicrobiae, which accounted for 87.13% of the diversity of the samples. The remaining classes were found in an average relative abundance of less than 5% (Figure 1B). When analyzed individually, the most abundant classes (>5%) in each type of sample were also Gammaproteobacteria, Bacteroidia, and Alphaproteobacteria, but in different relative abundances. Moreover, Polyaginia was also among the most abundant classes in organic lettuce samples (Figure 1B, Appendix A). In all types of samples, the bacterial class Gammaproteobacteria had the highest abundance. This class predominated in organic cabbage, lettuce, and strawberry samples, while in organic spinach samples its abundance was similar to that of Bacteroidia class.

The distribution at the genus level showed quite heterogeneous results. The most abundant bacterial genera (>2%) considering all samples were *Pseudomonas* (11.53%), *Flavobacterium* (7.23%), *Prosthecobacter* (4.64%), *Stenotrophomonas* (3.78%), *Pedobacter* (3.37%), *Cellvibrio* (3.29%), *Allorhizobium* (2.93%), *Sphingobium* (2.74%), *Massilia* (2.58%), *Achromobacter* (2.44%), an unclassified genus of the family Alcaligenaceae (2.26%), *Aquabacterium* (2.22%), and *Variovorax* (2.01%), which accounted for 51.03% of the bacterial FLA microbiome. Among these abundant genera detected as part of the microbiome of FLA isolated from fresh organic produce, *Massilia*, *Flavobacterium*, *Stenotrophomonas* and *Achromobacter* were also among the most abundant genera of the bacterial microbiome of FLA isolated from conventional lettuce samples [13]. Beta diversity analysis demonstrated significant differences in microbial communities among the various fresh organic produce types, emphasizing the complexity and variability of microbial interactions within FLA-associated ecosystems (Figure 2, Appendix A). *Pseudomonas* was the predominant genus in both organic lettuce and cabbage samples and the second most abundant genus in organic strawberry samples. However, this genus was not among the most abundant in organic spinach samples. The predominant genera in organic spinach and strawberry samples were *Flavobacterium* and *Sphingobium*, respectively.

The genera *Pseudomonas* and *Flavobacterium* have been previously described as ARB [34,35], and the genus *Sphingobium* has been previously found to be part of the FLA microbiome [18]. Moreover, in agreement with Moreno-Mesonero et al. [13], the genera *Massilia*, *Flavobacterium*, *Pseudorhodoferax*, *Stenotrophomonas*, and *Delftia* stood out among the most abundant genera in lettuce samples.

Among the identified FLA microbiome, some abundant bacterial genera not previously related to FLA were detected as part of the FLA microbiome: *Prosthecobacter*, *Pedobacter*, *Cellvibrio*, *Nannocystis*, *Polyangium*, and *Dyadobacter*.

Alpha diversity analysis was used to reflect the richness of bacterial communities detected as part of the FLA microbiome. Sample richness can be observed in rarefaction curves, which showed that the number of observed features in spinach samples was the highest, followed by lettuce, cabbage, and strawberry samples (Figure 3A). Faith’s Phylogenetic Diversity index, which indicates phylogenetic richness, showed a significant difference among the type of sample (Kruskal–Wallis test, *p* = 0.0012, Appendix A). Pairwise comparisons showed that the diversity of spinach samples was significantly different to that of cabbage (Kruskal–Wallis test, *p* = 0.0001) and strawberry (Kruskal–Wallis test, *p* = 0.0043) samples (Figure 3B, Appendix A). Beta diversity analysis indicated that there were significant differences in the bacterial communities of the FLA microbiome depending on the type of sample, both with and without considering the taxonomy’s relative abundance (unweighted UniFrac, Permanova test, *p* = 0.001, Figure 3C, Appendix A; weighted UniFrac, Permanova test, *p* = 0.001, Figure 3D, Appendix A). Pairwise comparisons showed that, when considering only the bacterial composition (unweighted UniFrac), all types of samples were different from each other. However, when considering both bacterial composition and its abundance (weighted UniFrac), all types of samples were different from each other (Permanova test, *p* < 0.05) except for the pair lettuce and cabbage samples (Permanova test, *p* = 0.053).

The study of the FLA microbiome of vegetables which are frequently consumed raw is of foremost importance to know their possible role in the potential transmission of bacterial human pathogens. In this concern, some of the identified genera in this study are of interest to public health because they contain species that are potentially pathogenic to humans. Many of the bacterial genera of public health importance which have been detected as part of the FLA microbiome in this work have been previously identified as ARB [36,37,38]. Out of them, *Pseudomonas* had the highest relative abundance. This bacterium has previously been linked to FLA and has been identified as an ARB [17]. *Pseudomonas mendocina*, identified in this work, is an opportunistic species with the ability to adapt to a wide variety of environments that causes nosocomial infections, most of them associated with hosts with compromised immune systems. The second relevant identified genus was *Flavobacterium*. This genus contains pathogenic species, among which *Flavobacterium meningosepticum* is found, which causes different nosocomial infections such as meningitis or endocarditis [39]. *Flavobacterium psychrophilum* is also a human pathogen that has been occasionally detected in water samples, biofilms, or river sediments that receive outflow water from infected fish farms [40]. Moreover, both *Pseudomonas* and *Flavobacterium* have been previously detected as part of the FLA microbiome isolated from conventional lettuce and water sources [13,17,18].

Other relevant genera found as part of the FLA microbiome in low relative abundances are *Acinetobacter*, *Aeromonas*, *Arcobacter*, *Bacillus*, *Brevundimonas*, *Klebsiella*, *Legionella*, *Mycobacterium*, and *Salmonella* (Appendix A). *Legionella* is one of the most studied genera associated with FLA [41]. Among its pathogenic species, *L. pneumophila* is a common ARB, causing severe pneumonia that can be fatal. In the present study, the genus *Legionella* represented almost 0.5% of the sequences, and even in two lettuce and one cabbage sample, it had greater abundance values (6.18% and 5.21% in lettuce samples and 3.56% in a cabbage sample). It has been previously demonstrated that this human pathogen can escape the digestion of FLA and even multiply inside amoebae such as *A. castellanii* or *V. vermiformis* [19]. The presence of *Aeromonas*, *Arcobacter*, *Legionella*, *Mycobacterium* and *Salmonella* inside FLA isolated from non-organic vegetables has been previously reported [9]. However, other genera which contain pathogenic species such as *Acinetobacter*, *Bacillus*, *Brevundimonas*, or *Klebsiella* have been identified for the first time as part of the FLA microbiome isolated from fresh organic produce in the current study, although they were previously found as part of the microbiome of FLA from water sources [17,18].

To identify and corroborate the presence of pathogenic species as part of the FLA microbiome, the sequence of each ASV of the relevant genera was confronted against the NCBI database using the BLAST tool. It was confirmed that ASVs contained some species classified in risk group 2, such as *Arcobacter cryaerophilus*, *Bacillus thuringiensis*, *L. pneumophila*, *P. mendocina*, *Salmonella enterica*, *Staphylococcus epidermidis*, or *Stenotrophomonas maltophilia*. That is, they could cause human disease, but there is generally little chance of posing a threat to laboratory personnel, the population, or the environment. Once pathogenic bacteria have been confirmed as a part of FLA microbiome, other specific studies to show their viability and infection potential could be addressed using other techniques such as PMA-qPCR (propidium monoazide-qPCR) or DVC-FISH (Direct Viable Count-Fluorescent in situ Hybridization) [13,42].

## 4. Conclusions

This is the first work in which the presence of FLA and their bacterial microbiome has been studied in fresh raw organic produce to highlight the critical need to comprehend the intricate relationship between FLA and bacterial communities in these produces. It has been demonstrated that FLA are ubiquitous in these organic vegetables, and the specific FLA *Acanthamoeba* spp. and *V. vermiformis* were detected in 65.0% and 25.0%, respectively, of the samples via qPCR. The bacterial microbiome of FLA was significantly different depending on the type of fresh organic produce. Some bacteria detected in this work, such as *Pseudomonas* or *Flavobacterium*, have been previously described as ARB; however, the presence of other bacteria which have been described as related to FLA for the first time in this work, such as *Prosthecobacter*, *Pedobacter* or *Cellvibrio*, underscore the complexity of these ecosystems and the need for comprehensive studies to unveil their significance. Moreover, among the FLA microbiome from fresh organic produce, human pathogenic bacteria, such as *A. cryaerophilus*, *B. thuringiensis*, *L. pneumophila* or *S. enterica* were detected, thus suggesting that FLA could be carriers of such potential pathogens and can act as transmission vehicles, possibly causing public health issues. Hence, it would be necessary to carry out further specific research (e.g., PMA- qPCR or DVC-FISH) to demonstrate the viability and infectivity of the different identified bacteria to determine the actual risk for consumers.

In conclusion, this study contributes valuable insights into the microbial ecology of FLA in raw organic produce. The implications of FLA as carriers of pathogenic bacteria necessitate continued research to elucidate the extent of their impact on food safety. Addressing these aspects is crucial for ensuring the health and well-being of consumers who rely on organic produce for its nutritional benefits.

## Figures and Tables

**Figure 1 foods-12-03102-f001:**
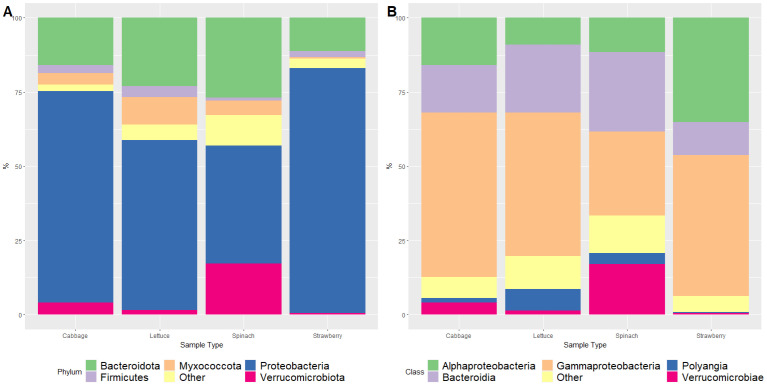
Relative abundance (%) of bacterial phyla (**A**) and classes (**B**) identified as part of the FLA microbiome isolated from fresh organic produce.

**Figure 2 foods-12-03102-f002:**
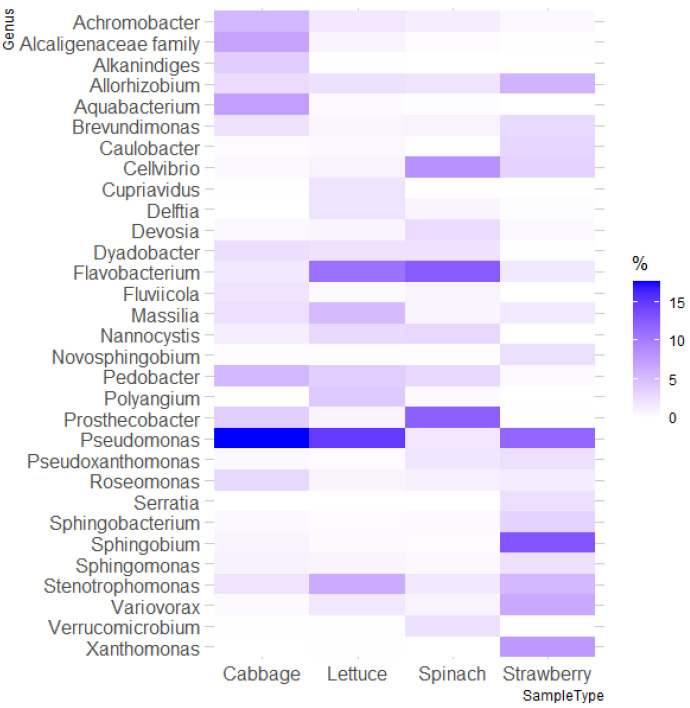
Heatmap of the relative abundance (%) of the most abundant bacterial genera (>2% in at least one type of sample) identified as part of the FLA microbiome isolated from fresh organic produce.

**Figure 3 foods-12-03102-f003:**
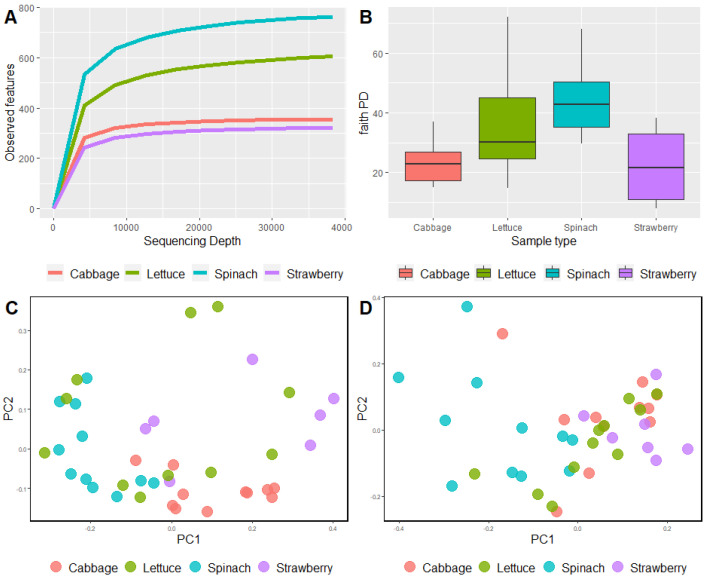
Microbial diversity analysis. (**A**) Rarefaction curve. (**B**) Faith’s Phylogenetic Diversity index. (**C**) Two-dimensional principal coordinate analysis (PCoA) plots based on unweighted UniFrac distance matrices. (**D**) Two-dimensional principal coordinate analysis (PCoA) plots based on weighted UniFrac distance matrices.

**Table 1 foods-12-03102-t001:** Detection of *Acanthamoeba* spp. and *Vermamoeba vermiformis* via qPCR in fresh organic produce.

	*Acanthamoeba* spp.	*V. vermiformis*
Lettuce	4/11 (36.4%)	1/11 (9.1%)
Cabbage	11/11 (100.0%)	4/11 (36.4%)
Spinach	4/11 (36.4%)	4/11 (36.4%)
Strawberry	7/7 (100.0%)	1/7 (14.3%)
Total	26/40 (65.0%)	10/40 (25.0%)

## Data Availability

The data presented in this study are available on request from the corresponding author.

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
