# Peer review of "Microbiome of Free-Living Amoebae (FLA) Isolated from Fresh Organic Produce: Potential Risk to Consumers?"

_foods, 2023, doi:10.3390/foods12163102_

Round 1
Reviewer 1 Report
Overall, this paper is novel and although the obtained results seem to be premature, they could provide interesting new tips to enrich the scientific knowledge on the food safety field. However, I believe that the following adjustments need to be taken into consideration.
The authors did a good job of synthesizing the literature, but the introduction could be ampliated with further details, for example could be interesting explain what is meant by organic agriculture products (L: 28) in opposition to the conventionally growth vegetables (L: 165); also, the topic of this work should be introduced in a broader way integrating the literature (e.g., 10.3389/fcimb.2018.00028).
Below I address some issues identify in the paper:
L47: Cryptosporidium and Giardia should be written in italic font.
L 49-50: the reference reports only a possible correlation between Legionella and FLA. It could be changed choosing another one that is fitting for FLA as bacterial predators.
L 61: Acanthamoeba and Vannella should be written in italic font.
L62: what type of vegetables treatments? They should be reported.
L 64-65: references? (e.g., doi.org/10.1016/j.clinmicnews.2010.11.001)
L73: Campylobacter jejuni, Legionella pneumophila or Listeria monocytogenes should be written in italic font.
L 86-87: describe what type of sterile bags were used and provide more details about the Page's amoeba saline (PAS) buffer.
L 100: has been done some microbiologic survey to verify the effectiveness of the sodium hypochlorite treatment in the killing of the bacteria non-internalized into FLA?
L 103: where was propidium monoazide (PAM) purchased?
L 169-171: reference?
L 179: should be explicit also in materials and methods section that outer leaves of the organic vegetable samples were selected.
L 204: i.e. should be written in italic font.
L 238-242: the sentence should be properly reworded.
In my estimation, this paper requires to improve the introduction adding additional information and clearly explain the lacking details in the materials and methods section.
Author Response
First of all, I would like to thank the reviewer for the reviews ans comments to improve the manuscript.
The authors did a good job of synthesizing the literature, but the introduction could be ampliated with further details, for example could be interesting explain what is meant by organic agriculture products (L: 28) in opposition to the conventionally growth vegetables (L: 165); also, the topic of this work should be introduced in a broader way integrating the literature (e.g., 10.3389/fcimb.2018.00028).
According with the reviewer recommendations, more details about organic agriculture has been added (L:38-43). Other literature has been also integrated (L: 64-65)
Below I address some issues identify in the paper:
L47: Cryptosporidium and Giardia should be written in italic font.
Done
L 49-50: the reference reports only a possible correlation between Legionella and FLA. It could be changed choosing another one that is fitting for FLA as bacterial predators.
According to the reviewer, the reference has been changed.
L 61: Acanthamoeba and Vannella should be written in italic font.
Done
L62: what type of vegetables treatments? They should be reported.
In the work of the reference cited, authors only mentioned washing and sanitation as a ready- to use lettuces. This information has been added.
L 64-65: references? (e.g., doi.org/10.1016/j.clinmicnews.2010.11.001)
The reference cited by the reviewer is now included. L: 81
L73: Campylobacter jejuni, Legionella pneumophila or Listeria monocytogenes should be written in italic font.
Done
L 86-87: describe what type of sterile bags were used and provide more details about the Page's amoeba saline (PAS) buffer.
According with the reviewer suggestions more details about bags and PAS buffer have been added. L. 101-111
L 100: has been done some microbiologic survey to verify the effectiveness of the sodium hypochlorite treatment in the killing of the bacteria non-internalized into FLA?
The hypochlorite concentration used is far above what is used to kill bacteria to make sure we kill bacteria outside FLA, according to Moreno-Mesonero et al., 2016 (ref. 22 in the manuscript).
L 103: where was propidium monoazide (PAM) purchased?
The information has been added. L:120
L 169-171: reference?
Now is right.
L 179: should be explicit also in materials and methods section that outer leaves of the organic vegetable samples were selected.
This information is included now in M and M.
L 204: i.e. should be written in italic font.
Done
L 238-242: the sentence should be properly reworded.
According with the reviewer suggestion, the sentence has been changed. L: 259-262
In my estimation, this paper requires to improve the introduction adding additional information and clearly explain the lacking details in the materials and methods section.
We have improved the introduction with all the suggestions of the reviewer.
Reviewer 2 Report
1. This is another report concerning FLA. Although this work is not particularly novel(organic vegetables) it is important and further adds to knowledge in this area.
2. It is critical that the authors explain how they have excluded bacterial DNA from the FLA. Where are the controls to show that this has been done? The lack of evidence that all external DNA has been removed is concerning.
3. The log reductions of FLA found following treatment with sanitiser is interesting and could be stressed more.
4. The authors state "Hence, it would be necessary to carry out further research to demonstrate the viability and infectivity of the identified bacteria to determine the actual risk for consumers." in the Conclusions. The authors should comment more on the limitations of qPCR.
Author Response
First of all, I would like to thank the reviewer for this review and comments to improve the manuscript.
- This is another report concerning FLA. Although this work is not particularly novel (organic vegetables) it is important and further adds to knowledge in this area.
As the reviewer comments is important to add knowledge in this area, and in this sense, this is the first work in which the presence of FLA and their bacterial microbiome is studied in raw organic fresh produce to highlight the relationship between FLA and bacterial communities in these produces.
- It is critical that the authors explain how they have excluded bacterial DNA from the FLA. Where are the controls to show that this has been done? The lack of evidence that all external DNA has been removed is concerning.
As reviewer says, the process by which external FLA DNA is excluded from the analysis is indicated in the text (lines 116-122). The evidence that external DNA is removed with this method has been optimized and reported in previous works of our team (doi:10.1016/j.resmic.2015.08.002 (Reference 22 in the text); doi: 10.1111/1462-2920.13856), therefore additional controls were not used in the assays of this study. L: 116-122
- The log reductions of FLA found following treatment with sanitiser is interesting and could be stressed more.
In the cited reference authors concluded that washing or rinsing lettuce leaves followed by spin-drying in a household salad spinner reduced the protozoan numbers only by a maximum 1 log unit.”
We have included some hypothesis for that so that there are no more references about this item. L: 189-192
- The authors state "Hence, it would be necessary to carry out further research to demonstrate the viability and infectivity of the identified bacteria to determine the actual risk for consumers." in the Conclusions. The authors should comment more on the limitations of qPCR.
We have not used qPCR for bacterial identification, only for FLA identification. So that when we state “, it would be necessary to carry out further research to demonstrate the viability and infectivity of the identified bacteria to determine the actual risk for consumers”, we mean specific techniques to identify viable bacterial genus such as PMA-qPCR or DVC-FISH. Now, we have specified that an added references in L: 342-346
Reviewer 3 Report
Although it is an interesting study, however, I found a major problem with the shallow depth of studies. There are a lot of problems in the manuscript particularly the structure/describing style and language of the manuscript. My individual comments are listed below:
The abstract needs a substantial revision. The aim of the study pointed out in the abstract is not clear and does not represent well the essence of the study. Furthermore, the abstract does not contain specific results expressed with values.
The introduction chapter is very generally written. Please provide an adequate background for the work objectives. There is no connection between the sentences. Consider discussing it deeper.
Line 73 - Please use the italic font and check the entire manuscript
The "Conclusion" section should be enlarged by including specific results and conclusions drawn from them, not only the general conclusion of the manuscript. Also, add future perspective in one to two lines..
References- Please check the format of the references. I recommend checking the guide for reference format.
The number of bibliographic sources is adequate, but less than 30% of the total bibliographic sources are from the last 5 years.
There are some grammatical errors and instances of badly worded/constructed sentences throughout the manuscript. Please refine the language carefully
Author Response
First of all, I would like to thank the reviewer for this review and suggestions to improve the manuscript.
Although it is an interesting study, however, I found a major problem with the shallow depth of studies. There are a lot of problems in the manuscript particularly the structure/describing style and language of the manuscript. My individual comments are listed below:
The abstract needs a substantial revision. The aim of the study pointed out in the abstract is not clear and does not represent well the essence of the study. Furthermore, the abstract does not contain specific results expressed with values.
The abstract has been revised and some information has been added or modified to improve it.
The introduction chapter is very generally written. Please provide an adequate background for the work objectives. There is no connection between the sentences. Consider discussing it deeper.
The introduction chapter has been improved.
Line 73 - Please use the italic font and check the entire manuscript
Done
The "Conclusion" section should be enlarged by including specific results and conclusions drawn from them, not only the general conclusion of the manuscript. Also, add future perspective in one to two lines.
The “conclusion” section has been enlarged according to reviewer suggestions.
References- Please check the format of the references. I recommend checking the guide for reference format.
References format has been checked.
The number of bibliographic sources is adequate, but less than 30% of the total bibliographic sources are from the last 5 years.
We have deleted some older references and added other from the last 5 years.
Comments on the Quality of English Language
There are some grammatical errors and instances of badly worded/constructed sentences throughout the manuscript. Please refine the language carefully
The language has been revised.
Round 2
Reviewer 2 Report
Thank you for your responses. I am generally content. Interesting area.